# Design of Hardware and Software Equipment for Monitoring Selected Operating Parameters of the Irrigator

**DOI:** 10.3390/s22093549

**Published:** 2022-05-06

**Authors:** Ján Jobbágy, Oliver Bartík, Koloman Krištof, Viliam Bárek, Roderik Virágh, Vlastimil Slaný

**Affiliations:** 1Institute of Agricultural Engineering, Transport and Bioenergetics, Faculty of Engineering, Slovak University of Agriculture in Nitra, Trieda Adreja Hlinku 2, 94976 Nitra, Slovakia; jan.jobbagy@uniag.sk (J.J.); xbartik@uniag.sk (O.B.); 2Institute of Landscape Engineering, Faculty of Horticulture and Landscape Engineering, Slovak University of Agriculture in Nitra, Trieda Andreja Hlinku 2, 94976 Nitra, Slovakia; viliam.barek@uniag.sk; 3Institute of Accounting and Informatics, Faculty of Economics and Management, Slovak University of Agriculture in Nitra, Trieda Andreja Hlinku 2, 94976 Nitra, Slovakia; roderik.viragh@uniag.sk; 4Department of Agricultural, Food and Environmental Engineering, Faculty of AgriSciences, Mendel University in Brno, Zemědělská 1, 61300 Brno, Czech Republic; vlastimil.slany@mendelu.cz

**Keywords:** agriculture, automation, irrigation, work quality, water, monitoring system, operation control

## Abstract

The aim of this paper was to design a device for monitoring the work of irrigation technology (in our case, irrigation by sprinkler). Two devices for monitoring selected irrigation operating parameters for two hose reel irrigation machines were designed. During the monitored period of connection of the equipment to the sprinkler, 15 irrigation doses were carried out for both sprinklers. Irrigation operating characteristics working pressure, hose reel speed and selected weather conditions temperature and humidity were monitored. When evaluating the results, we proved the need to monitor the operation of the sprinkler not only by the coefficient of variation *C_v_*, but also by introducing the coefficient of non-uniformity *a*. The results obtained indicate variability with respect to a particular irrigation dose and the applicable assessment method. The results were reviewed by one-way *ANOVA* analysis where observed coefficients and irrigation dose were considered as dependence factors. The results indicate a statistically significant impact of the applied quality coefficient of work and thus the impact of a particular device (*p* < 0.05, *F_crit_* = 2.77). When evaluating the effect of the included irrigation dose, we also showed a statistically significant effect in both facilities (*p* < 0.05, *F* = 1.92). By checking the operation of the hose reel irrigation machine, we managed to successfully apply the proposed classifications, which also perform the function of fault prediction. The proposed facilities show that proper plant operation and timely response can help create more efficient and sustainable irrigation services, not only saving water but also reducing costs for the owner.

## 1. Introduction

Given that there is currently a shortage of water, which is a global problem and growing every day, the issue of economic management of this resource needs to be addressed. Approximately 70% of the fresh water used in many parts of the world is used for irrigation, from agricultural fields to gardens [1]. Excess water that is not used efficiently causes negative effects on soils. In recent years, substantial water conservation has been a global goal. The sustainability of the system is currently directly linked to innovation, as people around the world are increasingly interested in climate change. This is primarily related to the lack of water, the deployment of the so-called clean energy and other challenges that require the development of new products and services that can lead to progress not only in the environmental and social dimension but also in economic terms [2]. In some countries, hose reel irrigation machines are still popular. Manufacturers in the sector are launching more efficient solutions built not only to achieve excellent irrigation but also to optimize features such as pumps, hose reels, hydraulic actuator systems and central processing unit control. Hose reel irrigation machines represent a category of machines that can be used to irrigate selected crops such as potatoes, seed maize, vegetables, and selected major crops (wheat, cotton, etc.) as well [3,4]. Juang et al., 2021, also studied the effects of irrigation on the relationship and yield of some crops. Their results show that when fields were irrigated, responses to crop yields (corn, wheat, soybeans, and cotton) became less sensitive to changes in precipitation and growing season temperature compared to responses observed in rainy conditions. Authors concluded, however, that irrigation and the presence of shallow groundwater levels (less than 3 m) may increase crop resistance to reduced rainfall and elevated temperatures, with effects depending on soil, texture, and soil organic content [5].

Monitoring hose reel irrigation quality of work and proper deployment, however, is one of the essential features, which we are convinced not only by us but also by other authors [4,6,7]. Irrigation by utilizing hose reel irrigation machines is constantly evolving and is one of the most widely used methods of irrigation. World agriculture is aimed at monitoring all components of the water footprint (an indicator of direct and indirect water consumption), because of which irrigation producers seek to optimize water supply (loss reduction) [4].

As already mentioned, among the solutions for spray irrigation, irrigation with hose reel irrigation machines have been used in practice, which is most often connected to hydrants with a flexible hose. The systems consist of components for mobile water distribution with a big gun sprinkler (combination with a stativ-travelers on an irrigation boom) or a bracket (located at the end of the hose). The hose is wound on a drum and in most cases, the result is an irrigated rectangle. For their function, they need to have pressurized water from the hydrant (pressure approx. 0.5–0.6 MPa). The principle of operation is therefore to slowly wind the hose on a spool using the most used turbine, hydraulics, electric linear motor, or hydrostatics with an independent combustion engine. To ensure even irrigation or programmed distribution of the required dose, the hose retraction rate must be continuously controlled by mechanical or hydraulic devices that often operate automatically [2,4,8].

Modern models are equipped with many automatic operations, including limiting the pressure drop in the turbine, stopping the winding at the end of irrigation, etc. [4]. To ensure the quality of the machine of smaller and larger dimensions, it is necessary to have a pumping station for the distribution of the necessary operating parameters [9,10,11].

The literature review shows that there are devices in the category of hose reel irrigation machines that have no electronic equipment (older and simplest devices), have monitoring of some functions (newer devices, e.g., non-contact measurement by magnetic sensors built into the gearbox [4,12,13]) or it is possible to monitor them with external devices (hose winding speed design and implementation were carried out at the Technical Faculty at SUA in Nitra in two versions, submitted as a utility model, more detailed information and tests of the device were realized and results in another publication [14]). Another addition to the model range is the control and adjustment of functions by microcomputers, the automatic unloading of the tripod and the insertion of supports when the irrigation is completed. The display shows the specific settings and the corresponding irrigation mode, the setting of functions such as pre-irrigation and additional irrigation. During operation, the computer also indicates the set winding speed, the associated current irrigation dose in mm, the remaining irrigation time, the length of the pulled-out part of the polyethylene hose. Some sprinklers also allow the connection of a small weather station to monitor wind speed and direction and the magnitude of precipitation [12].

The winding system is one of the most complex and most researched mechanical-hydraulic components because it affects the overall operation of the machine. The even distribution of water, the reduction of pressure losses, the setting of the winding speed of the travelers or the irrigation booms with precise control of the irrigation dose height depend on this modern system [13]. Until now, hose reel irrigation machines have been fitted with winders with a slow or fast return of the final irrigation water distributor with a working width of 30 to 75 m. However, the use of different nozzles allows the farmer to apply the most suitable spray and water spray for his plot and crop. Research experiments show that the use of the variable speed option brings a more even application of irrigation [15]. The advantage of travelers with big gun sprinklers is maximum versatility with the possibility of changing nozzles and height adjustment, to achieve the necessary spraying, depending on the properties of the soil and the stage of development of the crop. There is also an electronic sector setting deployment for this type. The result is subsequently the adjustment of the sector at a different overall working angle and at the same time its rotation relative to the longitudinal line of the hose which is wound on the spool [16].

The overview shows that research and development in the field of modernization are also being carried out in hose reel irrigation machines. A key component of the hose reel irrigation machine is the mechanism for extending and retracting the hose guide. It affects the uniformity of water distribution, load loss, precise regulation of hose retraction speed, folding of irrigation arms and the resulting accuracy of irrigation. Technological developments have begun to focus on better hose reel systems to increase efficiency by reducing the power required for operation and increasing the quality of work even at lower operating pressures. The modernization is a turbine with a choke perpendicular to the inlet with a built-in bypass, which achieves a reduction of load losses to a minimum and more precise control of a wide range of speeds [4].

The difference between the designed devices and commercialy available equipment is that commercial systems do not employ GPS coordinate monitoring systems and that parameter monitoring is provided on the basic design of the machine (not on the sprayer, e.g., pressure). Our solution is to create new and modern hardware and software equipment that can ensure sustainable irrigation, not only in spray irrigation conditions. The proposed system collects and analyzes the current machinery datasets and on the basis of those data, evaluates and predicts potential faults. This means that the quality of work does not depend on the quality of the GPS signal. The aim of our research was therefore to create a monitoring system that monitors the working activity of the sprinkler (function of system components, monitoring of humidity, temperature) and informs the operator about the specific operating state.

## 2. Material and Methods

The less researched area of irrigation concerning the use of informatization in irrigation technology inspired us for more attention. This is also due to reducing the overall water consumption and increasing the quality of work by monitoring the operating parameters of irrigation equipment. Research in this area has been going on for several years, where the development of the device has been gradually improved and the results of the initial versions have been gradually published in scientific journals. As already mentioned, an older version of the hose reel speed monitoring system was presented in a previous publication [14]. The very first proposed version only allowed the measurement of the hose winding speed with the need to monitor and manually record the results (year 2000). Later, an updated version (2012) allowed not only measuring the hose winding speed (via a tachometer) but also having the ability to write collected data (122 values at max.) to the memory of the microcontroller PIC16F877 (Microchip Technology Inc., Chandler, AZ, USA) [14]. Due to the expanding communication devices, the possibility of availability of an Internet connection at reasonable prices, the latest versions of devices have been designed and tested in recent years, which after many years of research have a more complex form and the ability to monitor via the Internet with fault prediction.

### 2.1. Hardware Design

Two hose reel irrigation machines (Z1 and Z2) were used for practical tests, where the designed device E1 (Figure 1 on the left) and the second device E2 (Figure 1 on the right) were used. The basic unit of the first designed device E1 is the Arduino Uno microcontroller (Figure 2 left, Arduino Uno Rev3, Arduino, No.30 Fushan South Road, Beilun, Ningbo, China, Distributor: GM Electronic SR) and the innovated second E2 is a more complex LilyGO TTGO base plate v1.4 (Figure 2 right, Shenzhen Xin Yuan Electronic Technology Co., Ltd., Shenzhen, 518,116 China). It contains a SIM 800L module (IMCom, Quad band, Shanghai, China) to support the monitoring of operating and weather parameters (pressure, position, humidity, and temperature) via sensors. Both microcontrollers have a very similar method of communication, where the version LilyGO TTGO v1.4 also contains a GSM communication module. The integrated GSM module communicates with the server via the GSM network and sends information about the device status to the server. The technical specifications are given in Table 1.

Arduino Uno is a microcontroller development board based on ATmega328p (Atmega328p Microchip, ATMEL, San José, CA, USA; operating voltage 5V, Input voltage limits: 6–20 V, flash memory 32kB, clock speed 16 MHz, LED_Built-in: 13), 14 digital input/output pins (of which six can be used as PWM outputs), six analog inputs, USB connection, power connector, ICSP interface and reset button, contains everything needed to operate a microcontroller, and connection to a computer via USB cable is easy [17].

The LilyGO TTGO v1.4 microcontroller (T-Call v1.4 ESP32) is a more modern development board equipped with additional components (SIM800L WiFi GPRS Module, Chipset: ESPRESSIF-ESP32 240MHz Xtensa^®^ single-/dual-core 32-bit LX6 microprocessor). The operating voltage is in the range of 2.7 to 3.6V with a working current of 70 mA [18]. To power the module from the battery only, it is necessary to use the EN button, otherwise, the module will not be powered from the battery.

GSM communication module SIM800L (Figure 3 on the left, as a separate device designed for the microcontroller, e.g., Arduino, SIM Com, Quad band GSM/GPRS module with support of frequencies 850/900/1800/1900 MHz) is a module with predefined telecommunication commands, which can be used dial phone numbers, receive calls, send, and receive SMS messages and enter the Internet. The internal logic of the SIM800L module works with a voltage of 3.7 to 5 V DC, the module is compact, and its dimensions are 25 mm × 23 mm. For the functionality of the module, it is necessary to connect the pin connectors marked 5V, GND, SIM_RX, SIM_TX and the SIM_RST pin is an optional pin for resetting the module.

In the second designed device, in which the LilyGO TTGO v1.4 microcontroller was used, the connection of the pin connectors is eliminated, the base plate manufacturer predetermined the communication pins and created the connection directly on said microcontroller. However, for proper operation it is necessary to define pin connectors programmatically (#define MODEM_RST 5, #define MODEM_PWKEY 4, #define MODEM_POWER_ON 23, #define MODEM TX 27, #define MODEM_RX 26). The communication between the data collection device using the SIM800L module and the server will be discussed in the following sections.

A standard pressure transducer (EARU electric, Figure 3 right, pressure-voltage transducer, Electro schematics [19]) with a 5 V DC supply was applied to monitor the line pressure. The working voltage range is in the range of 0.5 ÷ 4.5 V DC and the working pressure range is 0 ÷ 1.2 MPa (with an accuracy of ±0.25). In the case of connection for the Arduino UNO board, the measuring voltage is fed directly to the analog pin connector A0, which has a resolution of 10 bits. In the case of the LilyGO TTGO v1.4 board, which does not have analog inputs, it was necessary to connect the ADS1115 analog-to-digital converter (Figure 1 on the right, position 6, Figure 4 on the left), which has a resolution of 16 bits.

ADS1115 converter (ADC Bit rate 16-bit, number of channels 4, operating voltage 2.0–5.5 V, operating current: 150 μA) uses I2C bus (Adafruit Industries LLC, 150 Varick Street New York, NY 10013, USA for communication with LilyGO TTGO v1.4 microcontroller, United States, SN: 1517443417 [20]). The I2C bus is a multi-master computer serial bus that allows several sensors or transducers to be interconnected using two wires. The conversion of voltage to pressure is linear and a value of 0.5 MP corresponds to a pressure of 0 MPa. The maximum voltage value of 4.5 V corresponds to a pressure of 1 MPa. This means that a pressure of 0.6 MPa is represented by a voltage value of 2.9 V.

A combined DHT11 sensor (interface: Digital OneWire, Figure 4 stred, Keystudio Inc) was applied to monitor operating properties such as temperature (±2 °C accuracy) and humidity (±5% accuracy). The operating voltage of the sensor is in the range of 3.3 to 5 V with a temperature range of 0 to 50 °C and a humidity range of 20 to 90%.

This sensor communicates with the motherboard via the 1-Wire bus. 1-Wire is a slow data bus with low signaling and voltage. One characteristic feature of the bus is the possibility of using only two wires, namely data and grounding. 1-Wire devices contain an 800 pF charge storage capacitor to power the device while data transmission is active. Data come directly from the DHT11 sensor and are processed using the appropriate library and presented as ambient temperature and humidity.

The rotary encoder (KY-040, relative, Figure 4 right, Keyestudio, Shenzhen, KEYES DIY ROBOT co., Ltd., Shenzhen, China [21]) was used in our research to monitor the winding speed of the unwound hose to the spool. Another possibility would be to use it to monitor the position of a pivot or wide-area sprinkler. It works on the principle of two mutually shifted pulses, determining the distance and direction of rotation. The encoder works with a voltage of 5 V DC (or 3.3 V for the LilyGO TTGO v1.4 board). Two signals, generally called A and B, are used to determine the direction, and count the encoder speed. The principle of reading data from the encoder is illustrated by bitstream (Figure 5). The data coming from signal veins A and B are processed using a simple algorithm and converted to distance units. The advantage of a relative encoder is the simple construction and design of the sensor, the disadvantage is that such an encoder cannot store the position value after disconnection of the voltage. For the encoder connected to the sprinkler to work properly, the gear had to be dimensioned (Figure 6 on the right). In Figure 6 on the left, its mounting on the hose reel irrigation machines is shown.

For the proper functioning of the device, a suitable power source had to be designed, which, in both cases, consisted of a commercially available battery (lithium-ion battery with a working voltage of 20 V DC (Working voltage: 18.5–21.5 V) with a capacity of 4Ah, (Parkside, supplier: LIDL, SR, manufacturer: Kompernass GmbH, Burgstrasse 21, D-44867 Bochum, Germany) and the terminal designed by us (using a 3D printer, can be connected to a plug-in socket) for its connection to the device. The basic output voltage of 20 V is adjusted to 5 V DC using an LM2596 converter (Texas Instruments, 12,500 TI Blvd., Dallas, Texas 75243 USA) [22]. The battery is capable of operating in the temperature range from −20 °C to 70 °C, which is sufficient for general field conditions, voltage reductions, for example, using the resistance method or a combination of more complex electronic components. The available voltage dividers operate as standard with an efficiency of 50 to 90%, depending on the input and output voltage. Voltage transducers are used to convert a DC voltage of a different range to a DC voltage of another different range (most often 4–30 V DC). These converters are often used when the battery voltage is different from the required voltage of the DC element, motherboard, or appliance. These converters tend to be small and their outputs range from tens to hundreds of watts.

At the beginning of 2021, another important functionality was added to the data collection system—MessageCenter (MessageCenter, Figure 7)—for better communication and crisis prediction options. The upgraded version of the device running on the LilyGO TTGO v1.4 motherboard has a subsystem for checking the machine’s crisis situations (programmed to constantly check the database of error messages with the designed device). This means that it could intervene by sending a unique line in the event of a machine failure. This link contains the type of machine fault and the exact GPS coordinates. The machine’s crisis subsystem also has an informative character when it can tell the operator the approaching end of the irrigation cycle. The principle of the functionality of the MessageCenter subsystem consists of three points (1—sending a unique link to the server, 2—the link server processes and stores it in the database, 3—checking the database of error messages with the device).

This facility is usually located in the main building of the company near the access to the electricity and internet network. As soon as the device detects that there are unsent error messages in the database, it sends them to the specific operator to whom the irrigation machine has been assigned. Service assignment and management take place in the administration interface.

The allocation of equipment for the operator takes place in several steps. The first step is to create an attendant (as an individual) with an assigned phone number. In the next step, the operator is assigned a device to manage. One operator usually has more than one device, but it is possible to assign one device to more employees. This is done in the device pairing section, where you can assign, edit, and delete pairings.

### 2.2. Software Design

The software was designed and programmed on the open-source development platform Arduino (Figure 8). It is based on the C programming language. This platform includes the Arduino IDE development environment, which launches many communication, math, and sensor libraries that help programmers implement various interfaces, such as sensors, communication modules (Wi-Fi, Bluetooth), and the like. The Arduino IDE development environment is offered for all used Windows, Linux, and MacOS operating systems. The advantage of this platform is many freely available projects and solutions to programming problems, which the user can download and modify as needed. The sequence and state culture must be observed when writing the program.

The first lines of the program are devoted to the import of libraries needed for various peripherals, which are planned to connect to development boards of various manufacturers such as Arduino LilyGO TTGO and the like. In Figure 8, the first lines contain the command #include, which is used to insert the necessary communication libraries for sensors. Predefined functions are also applied (for the DHT11 temperature and humidity sensor the command was used: dht11.read (& Measured Temperature, & Measure Humidity, NULL)), through which the program reads the temperature and humidity from the sensor and then writes it to the variables Measured Temperature and Measured Humidity.

Another rule when writing a program is to define global variables and constants that are used throughout the program. The practical solution is to define them immediately after the import of the corresponding libraries, as it was solved in our case (import of the SimpleDHT.h library, connected sensor on digital input DI0, definition of the variable pin as int pinDHT11 = 0;) and then only work with it (pinDHT11).

After defining the header, the program consists of two main parts, setup, and loop. Setup is used for the initial setup of the program and is performed only for the first time after starting the microchip board. Loop (cycler) is the part of the program that is executed continuously while the microchip board is running. User-defined functions are written after the end of the loop. The programmer can thus define his own functions that need to be used several times in the program (Figure 9). In this case, function1 returns the sum of the input variables x and y.

#### Communication Interface and Server

Another important part of the proposed communication system (Figure 10) between the device and the user application for hose reel irrigation machines is the server, which at certain time intervals processes the data sent from the proposed device (E1, E2) mounted on the irrigator. The capacity of connected devices to the server is limited only by the computing power of the server.

The server has scripts used to authorize devices and process received data. The server also includes an administration interface for device management, display, and export of measured data.

The administration interface (Figure 11) is designed to be simple and intuitive, where after logging in, an overview of deployed devices in field conditions is displayed. It is possible to add others to the existing list of devices (right side of the main menu window). The main menu consists, as already mentioned, of an overview of devices, graphical and tabular results, the option to set up devices or notifications, and the option to log out. After activating the overview window, we will see a list of devices with an ID code, name, and activity indicator. Graphical and tabular data of results for individual devices allows the software to monitor within the selected date and time (e.g., the exact period). In addition, the program interface allows you to determine the average values of the monitored parameters (temperature, humidity, pressure, and winding speed). When selecting a table export, the results are available in five columns (ID—Record number, auto-incremental; SID—Specific device number; Datatype—data type, such as position or temperature; Data—column of measured data itself, and Timestamp—time stamp when the data were sent on the server).

When selecting under windows (device settings) from the administrator main menu, setting options are offered to the user (Figure 12). Different corrections can be selected for each device. Corrections are used especially if the operator finds that the measured data of the device does not correspond to the real data and the operator does not have direct access to the corrections of the device itself, he can also perform the correction in the administration interface. The subjects of corrections are:Inverted metering logic (+ = −);Distance constant (ratio of measured and required distance);Pressure tuning (±Bar);Temperature tuning (±°C);Humidity tuning (±%).

With these tools, the operator can react relatively quickly to device measurement inaccuracies. Communication and sending of data with the device take place at certain time intervals set in the device itself, except for crisis situations such as pressure loss or excessive speed change. Communication takes place on a secure SSL layer. The server authorizes the communication link, verifies its correctness, and processes it. The processed data are then stored in a database so that they can be presented to the operator. For the communication of the device with the server, a so-called REST API (Representational State Transfer) is an interface that can be used to create, read, and edit data using simple http calls (links).

Within the created hardware component (MessageCenter), software was also created for better communication and crisis prediction options. The types of output messages are listed in Table 2.

The upgraded version of the device running on the LilyGO TTGO v1.4 motherboard has a subsystem for checking the machine’s crisis situations (programmed to constantly check the database of error messages with the designed device). This means that it can intervene by sending a unique line in the event of a machine failure. This link contains the type of machine fault and the exact GPS coordinates. The machine’s crisis subsystem also has an informative character where it can tell the operator the approaching end of the irrigation cycle. The principle of the functionality of the MessageCenter subsystem consists of three points (1—sending a unique link to the server, 2—the link server processes and stores it in the database, 3—checking the database of error messages with the device).

### 2.3. Equipment Operation

Correct functionality of the device was secured by communication based on the GSM network, which is one of the most widespread standards for voice and data communication. The proposed device with the connection of hardware and software via the GSM network (with the possibility of data transmission) sends data to a server, which decodes them, verifies their correctness, and processes them (Figure 13). As already mentioned, the device mounted on the sprinkler is powered by an external source (battery), which lasts approximately 168 h for the connection to the Arduino UNO motherboard and 100 h of operation for the device using the LilyGO TTGO v1.4 motherboard. The available microcontroller materials show that the LilyGO TTGO v1.4 module is generally more energy-intensive because it uses a more complex control chip from Espressif. This version of the device with a newer microcontroller (LilyGO TTGO v1.4) was developed about a year later than the version with the Arduino UNO. The new version of the device has early warning and machine failure prediction systems. The system checks the accuracy of the data and the course of individual events. When the pressure drops or the machine stops prematurely, it immediately informs the operator via SMS with the type of machine fault and the current GPS coordinates of the machine. The Espressif control chip is used more during the constant data control than in the previous version of the device using the basic ATmega 328p chip.

The data acquisition device is programmed to send data to the server at a certain time interval, except for fault prediction, which does not follow a time schedule. The data are sent over a Secure Sockets Layer (SSL) and authenticated on the server using a unique key.

Table 3 shows an overview of selected differences between the two data collection facilities, both facilities were made and tested in field conditions on the same plot, on the same types of hose reel irrigation machines.

### 2.4. Practical Verification in Field Conditions

#### 2.4.1. Locality of Interest

The practical verification of the designed devices’ correct operation took place in the field conditions during the irrigation of seed maize. The irrigation dose realized by the hose reel irrigation machines was set to an average value depending on the winding speed of the hose on the spool. The irrigation dose represents the amount of water delivered per unit area in one uninterrupted time interval until the effective watering depth is reached. Figure 14 shows the area of interest, and the land on which irrigation took place falls under the area of the agricultural enterprise SLOV-MART, ltd., Kátlovce. The company manages a total area of about 2000 ha, of which 530 ha are irrigated. In total, the company has applied irrigation so far in three different ways, where they are already discharging drip irrigation in terms of economic burden (it was about 70 ha). In addition to drip irrigation, spray irrigation is used for crops (wide-area irrigation technology—320 ha and hose reel irrigation machines—140 ha). Deployment of wide-angle machines is more advantageous for large plots of land, especially without obstacles (saving time of human needs for given irrigation work). Obstacles, such as power grids or others, prevent the efficient use of wide-area sprinklers, in which case belt machines (smaller irrigation areas and greater time requirements of manpower) find application.

The lands covered by these enterprises belong to a warm, slightly humid climate area, which is characterized by a mild winter with January temperatures above −3 °C. Average annual temperatures range from 9 °C to 10 °C, while average temperatures during the growing season range from 15 to 16 °C. Temperatures below 0 °C start, on average, from 12.10. and end on 24 April. The multi-annual average atmospheric precipitation in the area ranges from 600 to 650 mm. For the growing season, the average total is 350–400 mm. The area is in a slightly humid area, which significantly affects the soil formation process. The increased amount of precipitation determines the shift of organic minerals in the soil. As a result of this process, a cadastre of brown earth with lying and rusty subsoil, popularly called “June”, was created on the ridges and flat plateaus. The slopes of the ridges are affected by water erosion, because of which the soils are heavily washed away. Cadastral soils are characterized by the following genetic types: brown soil, alluvial soil, and chernozem. The area is dominated by brown earth floodplain soils and belongs to a slightly humid area.

The goal of the agricultural company SLOV-MART, ltd., Kátlovce is, together with top technicians, ensuring and maintaining the functionality of irrigation equipment, during their maintenance, protection, repairs, and operation (reconstruction and repair of pipe networks and all water stations Trakovice, Žlkovce, Malženice 2, Malženice 3, Špačince, Oravné, Horné Dubové). When growing crops and providing the necessary amount of irrigation water, the company envisages improving the economic result and facilitating its restructuring and modernization, especially to increase its market participation and agricultural diversification.

#### 2.4.2. Applied Irrigation Technology

In this paper, we focused on the deployment of the proposed equipment on two Irtec hose reel irrigation machines (IRTEC HEAD QUARTERS, Via G. Mameli, Castelvetro di Modena, Italy). Selected plots were irrigated with hose reel irrigation machines, which were equipped with refurbished irrigation water distributors (Figure 15). The sprinklers were equipped with a mechanical system for controlling the irrigation dose and thus the hose winding speed.

The technical and operational parameters show that the total length of the Z1 hose reel irrigation machine hose is 380 m with a hose diameter of 90 mm. The range of the long-range sprayer is up to 35 m, i.e., with an overall irrigation width of 70 m. The operating pressure reached 0.59 ± 0.02 MPa. The total length of the hose at the second sprinkler Z2 was 350 m with a hose diameter of 82 mm. The range of the big gun sprinkler was also 35 m, and the working pressure reached 0.59 ± 0.01 MPa. 

The source was a water pumping station, which takes water from underground sources (Figure 16). Water enters the irrigators under the prescribed pressure through the pumping station (pumps, electric motors, filtration, and control room), pipeline, and hydrants. The technical parameters of the applied hose reel irrigation machines and the weather conditions for specific measurements are given in Table 4. At the temperature, the range and average values are given.

The Z1 and Z2 sprinklers in the basic version consist of a classic or tandem chassis, a hose wound on a spool, a control unit, and travelers with big gun sprinklers. The movement of the coil is ensured by a transmission and a turbine to which the pressurized water is led. These sprinklers are widely used not only in agriculture but also in irrigation of playgrounds and in domestic conditions. During practical tests, the settings of the spray angle stop (more oriented to one side) were changed for some hose reel irrigation machines, depending on the conditions in the field or obstacles contained (on the way, etc.).

A manometer and a flexible connection hose are available as standard. Since the available irrigation devices had different hose diameters, 90 mm for the Z1 sprinkler and 82 mm for the Z2 sprinkler, calibration of the water flow was performed to secure the same irrigation water volumes applied from both irrigation systems. The standard hose length is 400 m, but for specific machines, the hoses are shortened to 380 m (Z1) and 350 m (Z2) for practical reasons of maximum available irrigation distance, which was limited by field boundaries. These changes, however, do not affect the measurements or limit irrigation device operation parameters. The equipment is protected with safety panels as required and the sprayers are supplied with the prescribed nozzles. The upper (hose with a spool) is rotatable relative to the chassis and its components are galvanized. Some types allow hydraulic travelers to lift. In addition, a tachometer or control unit may be included. The turbine is directly connected to a gearbox with a built-in bypass. The speed is changed mechanically (if there is no control unit) using a four-speed transmission. The traveler is galvanized and mounted on four variable wheels. The device also has an output for the tractor Power Take-Off shaft (max. speed 540 rpm).

#### 2.4.3. Application and Testing of the Proposed Device

The plot on which the measurement took place is located near the village of Žlkovce, after which the Žlkovce 1 pumping station is named. Opposite the plot is the Malženice steam-water power plant. The proposed equipment was used on hose reel irrigation machines when irrigating seed maize. The quality of work of a hose reel irrigation machine is formed on the one hand by transverse (application of one of the methods and the coefficient of spray uniformity) and longitudinal spray uniformity, and on the other hand by the continuous hose winding speed and the correct value of operating pressure. Both designed devices were tested for deployment in field conditions for 16 calendar days under variable weather conditions. As already mentioned, the device was equipped with pressure (EARU), humidity (DHT11), and temperature (DHT11) sensors and a speed monitoring device (KY-040, reel-to-reel speed). The output will be graphical waveforms of the observed parameters. The average humidity values were for device E1 (58.97%) and for device E2 (57.29%).

As it can be seen from the available resources, standards, and methodologies, there are several evaluation options for determining the quality of work, that is, coefficients and degrees of uniformity or non-uniformity of spraying, resp. coefficient of variation. Because the device designed by us is focused on measuring the pressure and winding speed of the hose, we focused our efforts on observing the values of the coefficient of variation. An important factor was also the state of pressure, the results of which could be monitored at any time because the pressure gauges installed by the manufacturers proved to be unreliable. This condition could also be caused by the age of the machine.

To evaluate the quality of the work (in our case variability of hose winding speed and pressure variability), the methodology applied by Oehler [23] was used (not to measure winding speed, but the quality of spray uniformity), but also by other authors in evaluating irrigation quality ([24] spray non-uniformity was evaluated). The methodology is based on the calculation of the value of the average deviation *A* (average error of individual values of the winding speed from their arithmetic mean). From the achieved values of *A*, the value of non-uniformity is determined and according to the following formula (for our calculations in percent, the relationship was adjusted with a multiple of the constant 100):(1)$$a=\frac{A}{N_{m}}\cdot 100,\;\%$$

*a*—uneven winding speed, %

*N_m_*—average value, mm,

*A*—average deviation, mm.

The second method, to determine the quality of work unevenness, was used to calculate the coefficient of variation *C_v_* [25], the dependence of the standard deviation σ, and the average height of the investigated quantity hm (hose winding speed and working pressure, *h_m_* value %):(2)$$C_{v}=\frac{\sigma }{h_{m}}\cdot 100,\;\%$$

#### Evaluation of Results

Statistically significant differences are assumed from the applied equations for evaluating the quality of work of the designed equipment (E1 and E2) mounted on hose reel irrigation machines (Z1 and Z2). From the obtained results, demonstrable or unprovable changes within the applied designed equipment are evaluated. For this purpose, the statistical apparatus STATISTICA [26] was used, in which a one-factor *ANOVA* analysis was used to evaluate and compare the results of the coefficient of non-uniformity and variation Formula (3), but also to evaluate the impact of the included irrigation dose Formula (4):(3)$$y_{ij}=µ+C_{i}+e_{ij},\;mm$$
(4)$$y_{ij}=µ+F_{i}+e_{ij},\;mm$$

*y_ij_*—measured value,

*µ*—overall mean,

*C_i_*—the effect of the coefficient of non-uniformity and variation,

*F_i_*—effect of the serial number of the applied irrigation dose,

*e_ij_*—random error with mean 0 and variance σ2.

Finally, an overall evaluation of the achieved results was performed.

## 3. Results and Discussion

As part of our long-term experience in the Department of Machinery and Production Biosystems (Institute of Agricultural Engineering, Transport and Bioenergetics; Faculty of Engineering; Slovak University of Agriculture in Nitra, Nitra, Slovakia), we managed to gain a lot of experience and thus design equipment for monitoring selected weather conditions and operating parameters of a sprinkler. In the paper, two devices E1 and E2 were tested on two different hose reel irrigation machines Z1 and Z2. Since we aimed to test the functionality of the proposed monitoring equipment and, on the other hand, the quality of the work of the irrigators, we had an agricultural plot on which seed maize was grown (Figure 16). In the methodological part of the paper, it was stated that the most important feature of the latest version of the device is, in addition to the function of monitoring selected parameters, also the function of fault prediction. The method of communication is based on the GSM module that communicates with the server. A more detailed description of data transmission and communication is described in the methodological part of the paper. The designed devices monitor the real-time data that are backed up and send fault prediction messages in the event of a fault. The monitoring equipment was in working condition in combination with the sprinkler for 16 days during the irrigation season. The values of the selected weather parameters (temperature and humidity) were obtained in the set time intervals (10 min), then they were sent to the server created by us directly from the device (www.irrigationbugs.com; accessed on 21 March 2021).

The implementation of the system made it possible to collect enough of the obtained results for the subsequent assessment of the quality of work of the hose reel irrigation machine and the monitored variability of information on weather parameters during irrigation.

Graphic (Figure 17—Z1 devices and Figure 18—Z2 devices, the graph also contains sensitive data about a specific measuring device, which we filtered out for blurred reasons—blurred), and tabular (Table 5—one example in the form of export from the administration interface) results data for both devices, allowing the software on the PC or tablet to monitor within the selected time (date and time, or the exact time period). From the obtained monitored data, it is possible to assess the change in operating and weather parameters. At the bottom of the table, the export contains data on average temperature, humidity and pressure, or winding speed, calculated from the exported data. The graphical representation shows the course of reducing the length of the pulled-out hose (traveler distance from the spool) of one applied irrigation dose, the time of the device transfer, and the partial section of the next irrigation dose. The working pressure values were also zero when the sprinkler was moved. However, the monitoring of selected weather conditions did not depend on the implementation of irrigation, but only on the connection of the proposed equipment E1 and E2 to the energy source. The results show that within the investigated period of deployment of the proposed equipment, it is possible to filter the necessary time periods and obtain detailed information on selected weather and operating conditions, not only after the end of the work but also during it. The advantage, as pointed out, is also obtaining information about irrigator failures (pressure or winding).

As already mentioned in the methodological part, we managed to successfully deploy in the system an innovated version of the device running on the motherboard LilyGO TTGO v1.4, a data collection device—MessageCenter (for better communication and crisis prediction capabilities). In the event of a fault that occurred during the tests, an SMS message was sent to the mobile device (sending a unique line). This link (thus also SMS) contained the type of machine failure and the exact GPS coordinates. The machine’s crisis subsystem also has an informative character when it can tell the operator the approaching end of the irrigation cycle. An example of a message is shown in Figure 19 (message: device speed fault, GPS coordinates of the device position are also listed for the message). The functionality of the entire monitoring and fault prediction system consisted primarily of sending a unique line to the server. In the second point, this link server processed and stored it in the database. In the third point, the MessageCenter device itself is present, which is built on the LilyGO TTGO v1.4 motherboard and programmed to constantly check the database of error messages with the data collection device. The list of administration interface faults in the MessageCenter section is shown in Figure 19. Another message was a pressure failure (if the pressure drops below 3 MPa, a message about a pressure failure in the system is sent).

The program interface allows you to add operator identification data, telephone numbers, pair the device with telephone numbers, and then inform the irons about the specific status of the device. The devices were therefore successfully set up and the data were sent to the two telephone numbers we had chosen.

In the tested period, which lasted 16 days, irrigation took place with several Irtec hose reel irrigation machines (technical parameters are given in the methodological part of the paper), of which two irrigators were subjected to quality testing. The speed of winding the hose on the spool was evaluated, and the values were calculated from monitoring the wound distance in 10-min time sequences. The working position of irrigators (both) changed 15 times during the monitoring period. Irrigation time periods depended on the set irrigation dose (related to the speed of winding the hose on the spool), they ranged from 11 h to 16 h. The value of the operating pressure was monitored in the values of 0.5 to 0.6 MPa, the pressure failure occurred only once, and we removed it. The evaluation of the quality of work was based on the determination of the proposed coefficient of non-uniformity and the coefficient of variation for both proposed devices E1 and E2. Figure 20 presents the results of the work quality coefficients and *C_v_* of the selected hose reel irrigation machine Z1 at 15 irrigation doses realized in succession. The results show that the results of the coefficient of variation *C_v_* vary from 14.26 to 52.20% and the coefficient of non-uniformity and from 11.69 to 42.45%. Figure 21 shows the results of the variability of the stated quality coefficients of work at the second hose reel irrigation machine (more modern monitoring equipment, E2). The results show that the results of the coefficient of variation *C_v_* vary from 27.23 to 62.29% and the coefficient of non-uniformity and from 21.80 to 49.74%. From the above, it can be concluded that with the reliability of monitoring the results of work designed by the equipment E1 and E2, the quality of hose reel irrigation machines was different. Based on the achieved measured and graphical results, statistical verifications of the significance of the impact of the selected hose reel irrigation machine and the coefficient of quality of work, on the one hand, and the impact of the included irrigation levy, on the other hand, were also performed. *ANOVA* one-factor analysis was used for statistical evaluation of the results, where the results showed a statistically significant influence of the applied quality coefficient of work and thus also the influence of a specific device (*p* < 0.05, *F_crit_* = 2.77). When evaluating the effect of the included irrigation dose, we also showed a statistically significant effect in both facilities (*p* < 0.05, *F* = 1.92). This means that when all coefficients and methods are introduced, there is a statistically significant difference between the examined coefficient values between at least two evaluation methods. If the null hypothesis is rejected, the alternative hypothesis comes into play. This means that not all mean values are the same (at least one of the mean values is different from the others).

In addition to operating parameters, the proposed equipment helped to monitor weather conditions (humidity and temperature). In the future, it is planned to expand the equipment with the possibility of a power supply with an energy source connected to the solar panel. The endurance of the energy source will thus enable the monitoring device of the quality of the irrigator’s work during the entire irrigation season. Through the proposed devices, irrigators were notified in good time of a pressure failure via a fault prediction function. They were able to monitor the progress of the irrigators and thus take measures to improve them.

The plant was designed over several months to years (taking the initial proposals into account), with research and calibration taking place during the irrigation season and the growing season of the crop. The value of the set irrigation dose depended on the decade of the growing season and affected the speed of winding the hose (in other words, the time required to wind the individual sections of the total length of the pulled hose). In addition to this investigated property, the measurements also focused on the working pressure and selected weather conditions (humidity and air temperature).

There are many publications focused on monitoring the issue of irrigation that deal with irrigation in terms of monitoring evapotranspiration [27], the amount of water required and water quality [28], etc. [29]. However, many of them pursue the goal of reducing water consumption for a given area and carry out processes for improving the quality of work and computerization [30].

For example, in Pakistan, surface irrigation methods are commonly used to grow crops to meet the demand of an ever-growing population [31]. Practical methods of water application in agriculture usually lead to excessive water losses due to uneven water application, that is, places of excessive irrigation. This is also the reason for the deployment of more modern and economical equipment. Sprinklers in combination with a hose reel irrigation machine represent one of the methods of applying irrigation water, which can bring higher efficiency [32]. However, these facilities must be properly designed and then properly managed in different site-specific conditions. The primary requirement is still the fact of increasing the quality of work and thus achieving the right performance [6].

Application and distribution uniformity (main monitored parameter—operating pressure) was determined by standard evaluation methods. The authors found that the application efficiency and uniformity of distribution of the hose reel irrigation system vary from 66 to 74% with a corresponding base pressure range of 0.38 to 0.46 MPa. These irrigation systems are suitable for all types of land and small plots, are easy to move and operate, and are cost-effective from an economic point of view. Therefore, surface irrigation methods are commonly used to grow crops to meet the demand of an ever-growing population [31,33]. The available sources of several research works show that the quality of work, that is, the coefficient of spray uniformity, is given mainly by weather conditions (uncontrollable wind direction and speed) but also by design system variables (sprayer brand, nozzle size and type, and sprayer pressure and spacing) [6,9,32].

In recent years, modern irrigation technologies have come to the forefront of agricultural development for water conservation; these are highly efficient and crucial in the absence of agricultural water resources [8,34,35].

The essence of today’s modern irrigation system is to reduce water consumption and thus ensure the needs of regional sustainable development and local conditions [28,35]. The deployment of computerization and precision irrigation helps to reduce water consumption, and water is applied in adequate quantities in sectors or zones [36]. Glória et al. [37] stated that the conservation of natural resources is currently a growing problem, and that water scarcity is a reality that occurs in several parts of the world. One of the main strategies used to combat this trend is to use new technologies. An automatic irrigation management system for agricultural fields has been developed with a wireless network of sensors and actuators. Tracking variability through a mobile application today facilitates the work of irrigators. Agriculture is essential for the population and is a major consumer of water (more than 70% of the world’s fresh water).

This is a global problem and water-saving measures and scenarios need to be developed to sustain production for a growing number of consumers and water shortages [38]. Sustainability and the use of technology in agriculture help to alleviate these problems. With the development of technology and the application of the necessary information in our daily lives and activities, farmers can very easily and quickly access the necessary data [39]. Other authors also agree that globally, crop irrigation is the largest consumer of fresh water. Water scarcity is growing worldwide, leading to stricter regulation of its use in agriculture. This requires the development of irrigation practices that are more efficient in the use of water but do not jeopardize crop quality and yield. The introduction of irrigation in the exact dose, time, and place required helps these trends. Additional technologies and computerization need to be integrated into the agricultural process to maximize the benefits of irrigation [8,30,40]. Moreover, Borsato et al. [41] stated that water savings can be achieved by improving on-farm irrigation efficiency. To achieve this, it is necessary to choose the most appropriate irrigation technology, which, however, is often not an easy process [42].

From the research, the achieved results, and their comparison in terms of environmental, economic, and energy performance of irrigated and non-irrigated crops are therefore important. The output should be a sustainability analysis of irrigation practices within different irrigation systems. The sustainability of irrigation systems can be assessed by water-related indicators (water use efficiency, irrigation water efficiency), crop (crop growth rate, harvesting index, etc.) or energy and economic indicators (energy consumption, power and energy costs, and water [10,36,37]). In addition, the solution we have proposed has been tested in a seed maize stand, however, its application is not limited to this crop but is also applicable to other types of stands and crops [5].

Various conclusions lead from various sources aimed at increasing the quality of irrigators’ work, introducing informatization and monitoring not only the operational characteristics of machinery but also weather conditions [8,43,44]. Deploying the proposed equipment in practice will help irrigators to increase the quality of work and save such an important gift of land as water.

## 4. Conclusions

The paper paid attention to the design of two devices for monitoring the most important operating characteristics of the sprinkler and selected weather conditions. The result of the research was the design of hardware and software equipment that are improving the field operation of selected irrigation technology. Designs were tested on hose reel irrigation machines with possibilities to be applied to other types of irrigation technologies after adaptations. By suitable modification of the selected component (gear) and subsequent calibration, it is possible to use the designed equipment for wide-area irrigators. A highly valued benefit was the fitting of the proposed equipment into practical conditions. The facilities informed the irrigators about the current values of the activities of the hose reel irrigation machines, or about the specific monitored failure. The results of this new study show that the use of the obtained real-time weather data is a great advantage. It was also possible to understand that is only the state of the environment but also the operating parameters increased the efficiency of irrigation.

The results show that the hypotheses concerning the significant impact of the used irrigator or the applied coefficient of quality of work have been confirmed to us. In the case of confirmation of the second established hypothesis, concerning the order of the applied irrigation benefit, the results again showed a significant effect. So far, we have not made any publications that focus on the design of equipment for monitoring hose reel irrigation machinery speed and working pressure. The difference in this study in the evaluation of the quality of work of hose reel irrigation machines was also in the application of a new parameter, the irrigation non-uniformity coefficient. Another difference from commercial equipment is that parameter monitoring is provided on the basic design of the machine. Our solution was to create new and modern hardware and software equipment that can ensure sustainable irrigation, not only in spray irrigation conditions. The application of the equipment created by us allows irrigators to respond in a timely manner at a precisely determined place, and thus reduce the excess irrigation dose, respectively there is no state of hose reeling without irrigation.

In many agricultural companies, reel hose irrigators without modern electronics are still used, in which case the operator will later find out the current state of the irrigator or the event of a failure. Depending on the operator’s arrival delay to the device and in the event of a failure (assumed in our case) of winding the hose on the spool, the set irrigation dose is still applied to the same spot. A given irrigation dose would clearly increase water consumption, and thus economic costs, which were not quantified in the paper. This issue is a topic of future research, pointing out the overall losses that could occur at different time intervals for the operator’s arrival at the irrigator. If these devices are used, the operator can react immediately and react in time based on fault prediction.

## Figures and Tables

**Figure 1 sensors-22-03549-f001:**
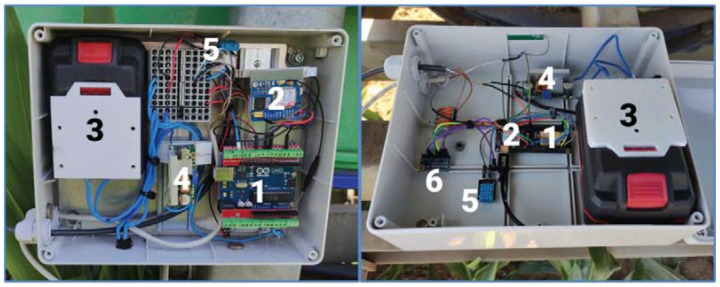
Designed devices E1 and E2 for sprinklers, (E1-left, 1-Arduino Uno, 2-module SIM, 3-source with terminal, 4-voltage divider, 5-sensor DHT11; E2-right, 1-Arduino LilyGO TTGO v1. 4, 2-module SIM, 3-source of power supply, 4-voltage divider, 5-sensor DHT11, 6-ADS1115).

**Figure 2 sensors-22-03549-f002:**
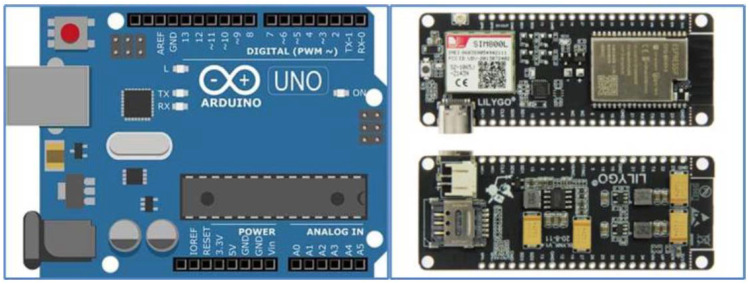
Arduino UNO and LilyGO TTGO V1.4 module.

**Figure 3 sensors-22-03549-f003:**
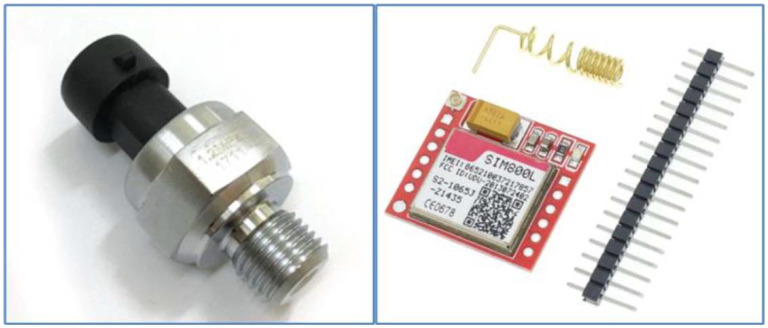
SIM800L communication module, EARU Electric pressure sensor (0–0.1 MPa, 0.5–4.5 V DC).

**Figure 4 sensors-22-03549-f004:**
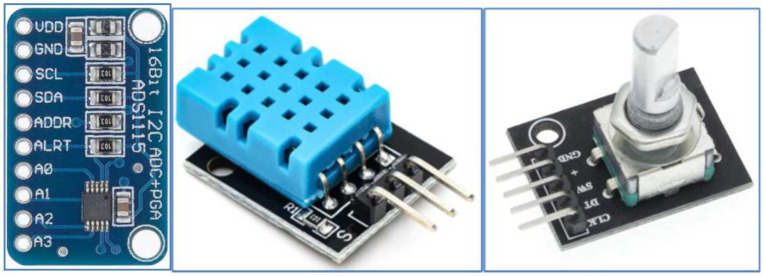
Analog to digital converter ADS1115, Combined sensor DHT11, Rotary encoder.

**Figure 5 sensors-22-03549-f005:**
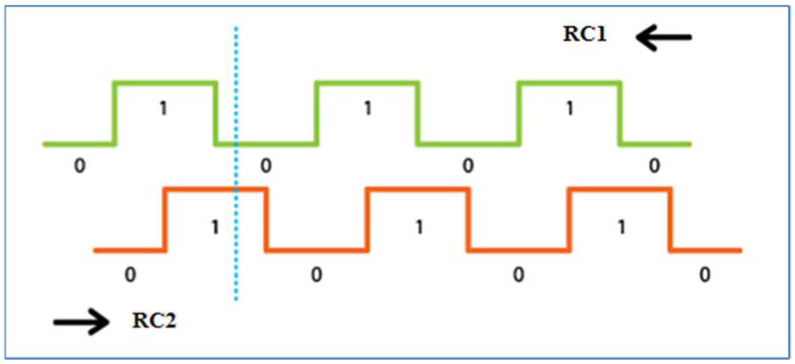
Encoder operation principle, RC1—counter clockwise, RC2—clockwise.

**Figure 6 sensors-22-03549-f006:**
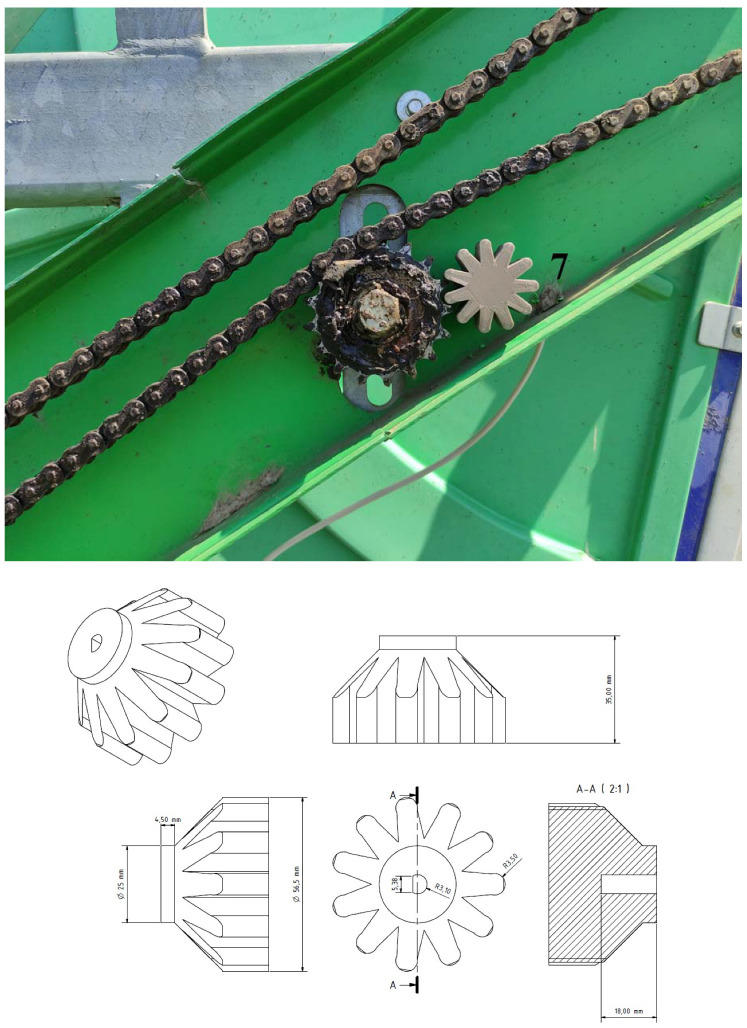
Designed gear, sprinkler seat.

**Figure 7 sensors-22-03549-f007:**
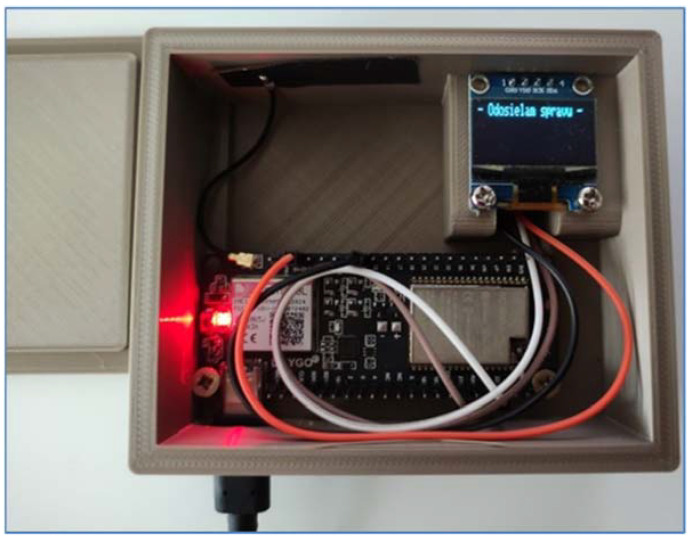
MessageCenter.

**Figure 8 sensors-22-03549-f008:**
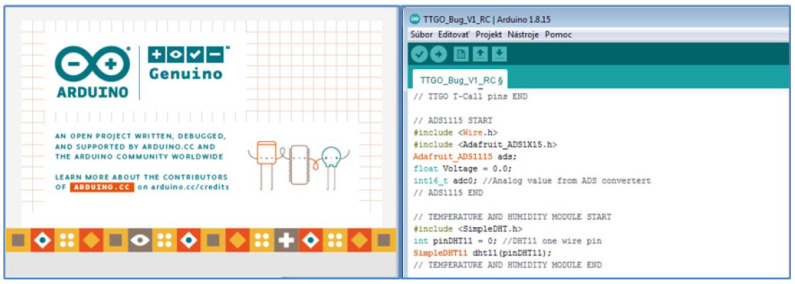
Working windows of the programming language.

**Figure 9 sensors-22-03549-f009:**
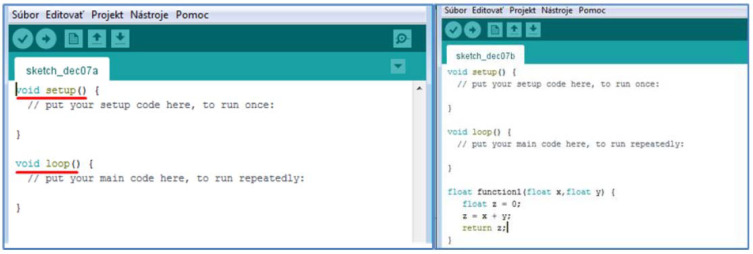
Header definition, setup, and loop.

**Figure 10 sensors-22-03549-f010:**
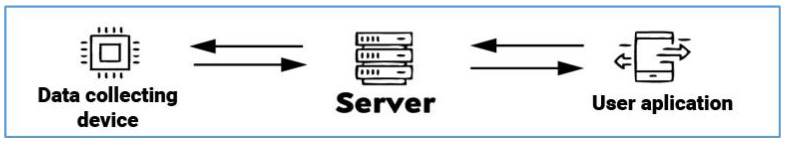
Simplified communication scheme of the data collection system.

**Figure 11 sensors-22-03549-f011:**
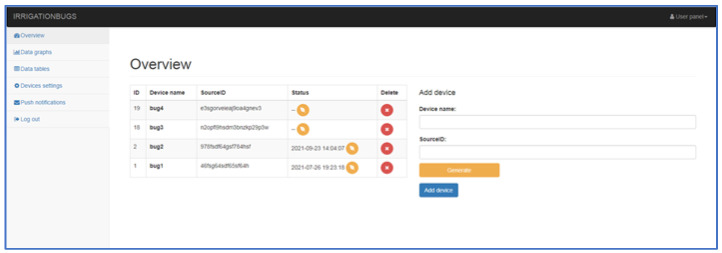
Home page of the administrator of all installed devices, 1—overview, 2—list of the machine, 3—add to machine.

**Figure 12 sensors-22-03549-f012:**
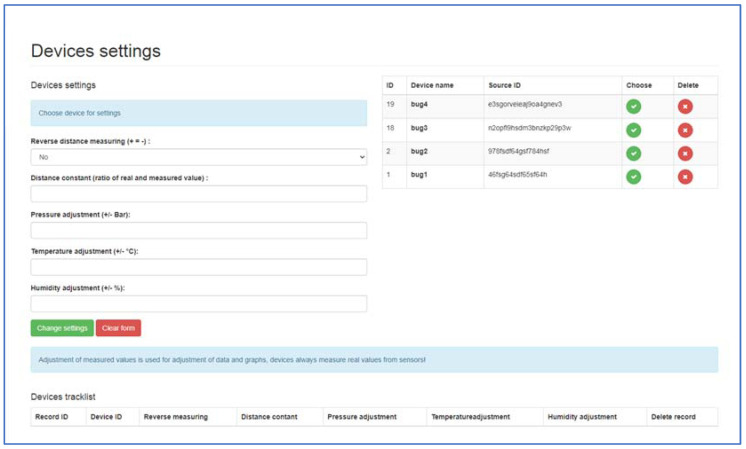
Menu—Device settings, 1—view for setting of the machine.

**Figure 13 sensors-22-03549-f013:**
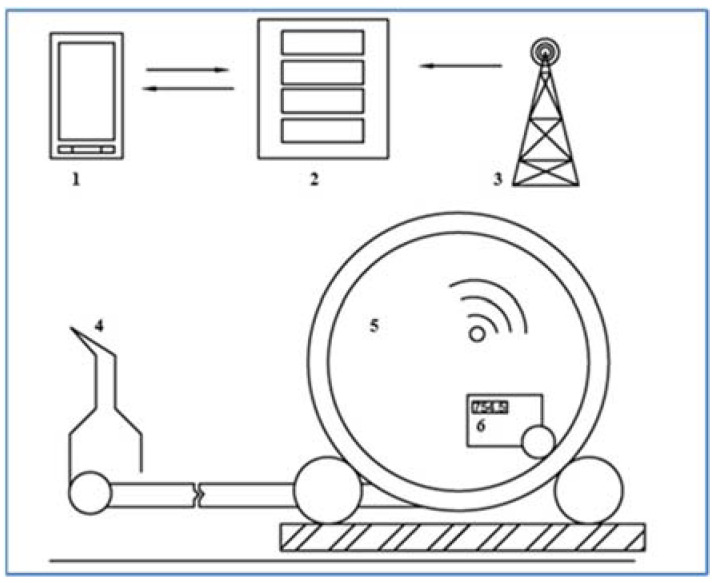
Block diagram of the principle of communication between the device and the smart device: 1—mobile smart device, 2—server, 3—signal GSM tower, 4—big gun sprinkler, 5—hose reel irrigation machine base frame, 6—data collection device.

**Figure 14 sensors-22-03549-f014:**
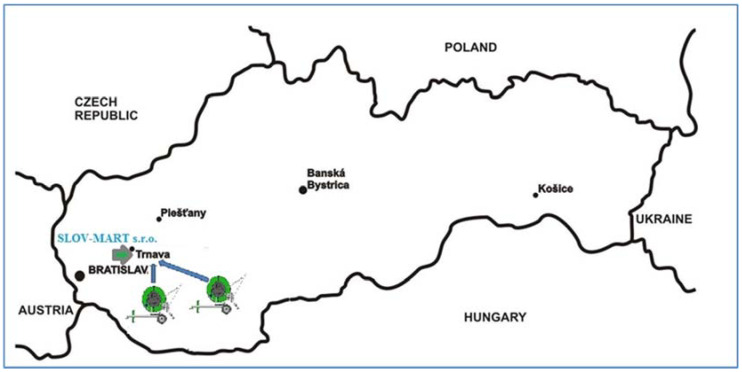
Location of measurement plain on western region on Slovak Republic map.

**Figure 15 sensors-22-03549-f015:**
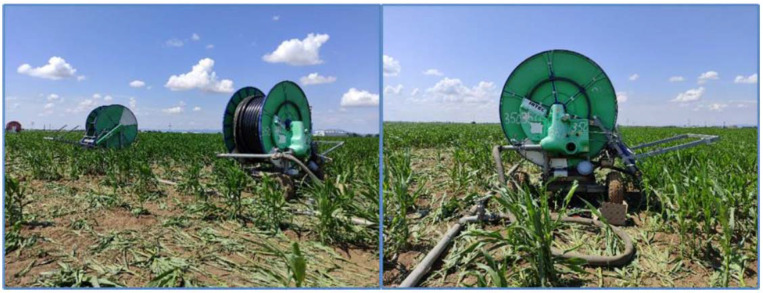
Application of hose reel irrigation machines (Z1 and Z2).

**Figure 16 sensors-22-03549-f016:**
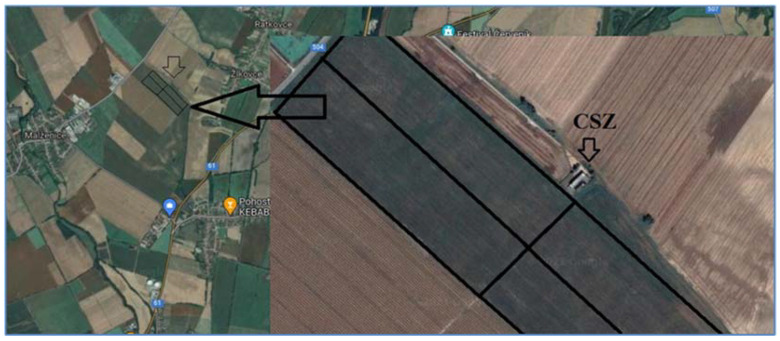
Locality of irrigation installation, CSZ—water station.

**Figure 17 sensors-22-03549-f017:**
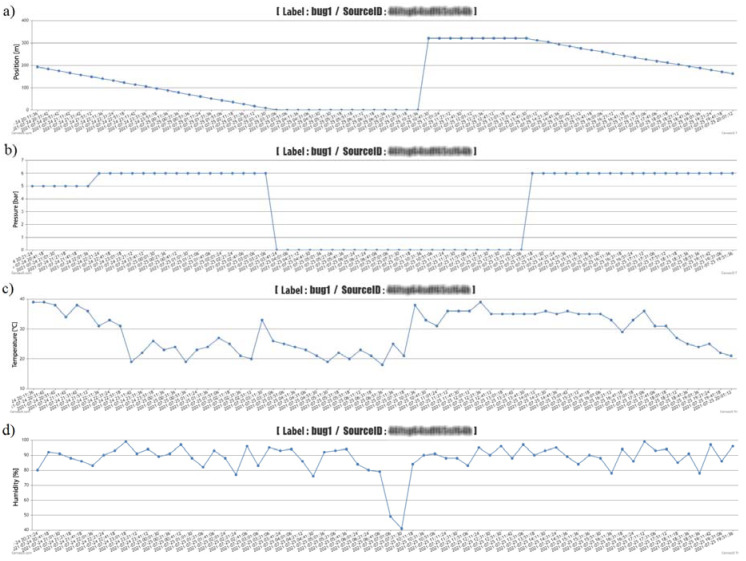
Graphic display of measured data from the administration interface—device Z1, (**a**)—pressure, (**b**)—distance, (**c**)—humidity, (**d**)—temperature.

**Figure 18 sensors-22-03549-f018:**
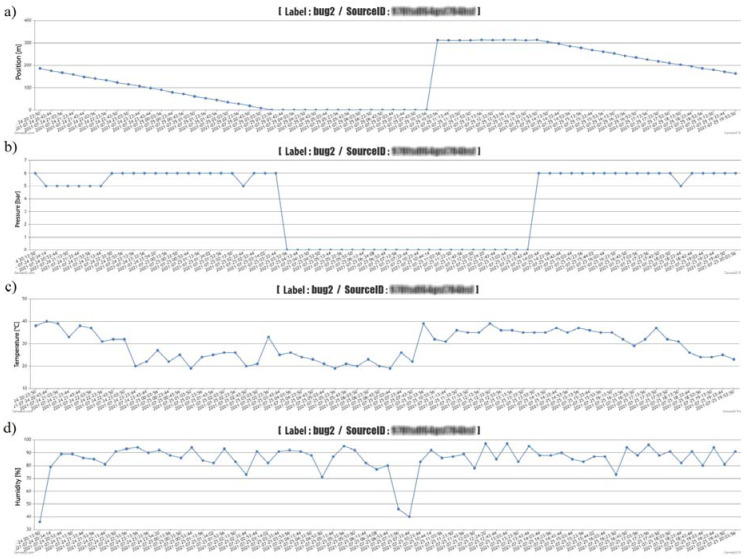
Graphic display of measured data from the administration interface—device Z1, (**a**)—pressure, (**b**)—distance, (**c**)—humidity, (**d**)—temperature.

**Figure 19 sensors-22-03549-f019:**
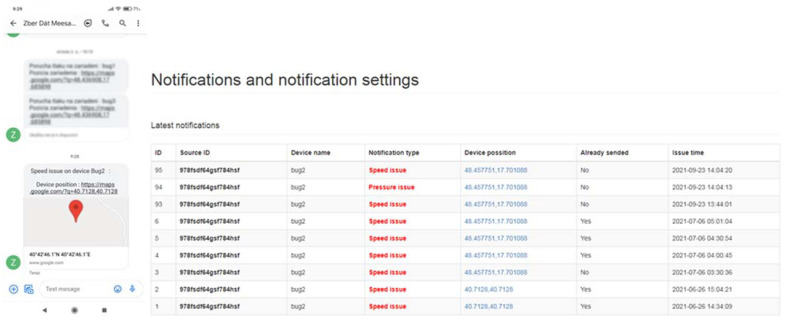
Preview of message sent via MessageCenter, list of faults in the administration interface, 1—device name, 2—type notification, 3—distance of device, 4—send, 5—report time.

**Figure 20 sensors-22-03549-f020:**
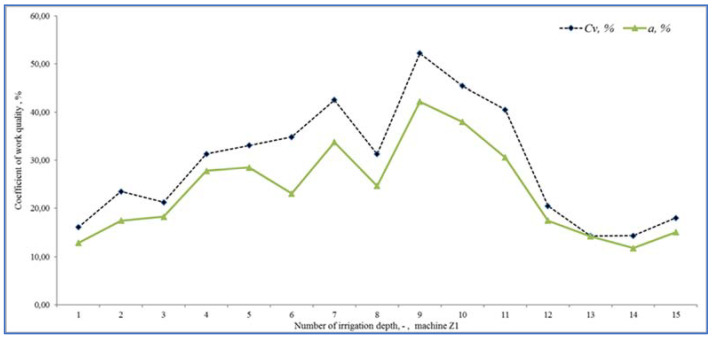
Coefficient of variation and non-uniformity and *C_v_* (%) of 15 different irrigation doses (Z1).

**Figure 21 sensors-22-03549-f021:**
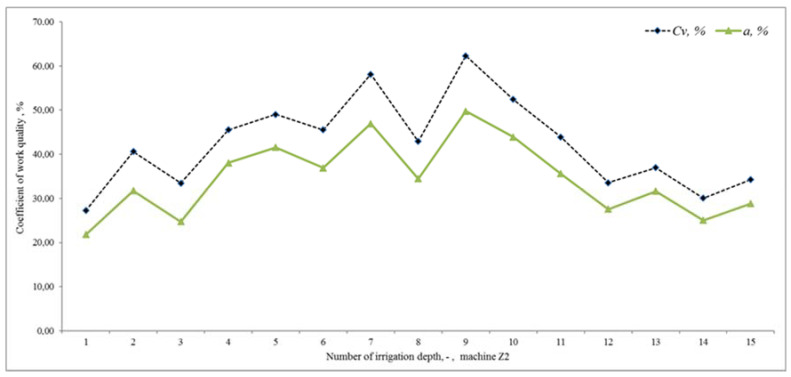
Coefficient of variation and uniformity and *C_v_* (%) of 15 different irrigation doses (Z2).

**Table 1 sensors-22-03549-t001:** Technical and operating parameters of sensors.

Pressure Sensor, EARU	
Working voltage, DC, V	5 ± 0.25
Working temperature, °C	−40~125
Storage temperature, °C	−40~130
Output type, V	Voltage output, 0.5~4.5
Range ability, Mpa	0~1.2
Overload pressure, Mpa	2.4
Explosion pressure, Mpa	3.0
Temperature and air humidity sensor, DHT11	
Humidity range, RH, %	20~90 ± 5
Temperature range, °C	0~50 ± 2
Working voltage, V	3.3~5
Resolution: Humidity %, Temperature °C	1 and 1
Current supply, mA	0.5–2.5
Sampling period, s	1
Rotary encoder	
Number of pulses for 360°	20
Dimensions, mm	31 × 19 × 29
Weight, g	8,4
Working voltage, V	5 ± 0.10

**Table 2 sensors-22-03549-t002:** Output messages from MessegeCenter.

NoP, Information	Type of Reports	Message	Description
1 (Error)	Pressure error	XY Device Pressure Failure, Device Position https://maps.google.com/?q=long,lat (accessed on 21 March 2021)	Device response time 1 min, appears every 30 min until fault 2 is rectified
2 (Error)	Flow error	Speed error on XY device, Device position https://maps.google.com/?q=long,lat (accessed on 21 March 2021)	The reaction time of the device is 1 min, it appears every 30 min until the fault is rectified
3 (Warning)	Low battery	XY device battery is low, Device position https://maps.google.com/?q=long,lat (accessed on 21 March 2021)	Reaction time of the device 1 min, appears 1 time
4 (Information)	The device has finished	XY has finishedDevice position https://maps.google.com/?q=long,lat (accessed on 21 March 2021)	The reaction time of the device is 1 min, it appears 1 time at the end of the watering cycle

NoP—number of priorities; XY—common name of the facility; long, lat—longitude and latitude.

**Table 3 sensors-22-03549-t003:** Operating and technical parameters of the proposed equipment.

Parameter	Designation, Value	
Equipment	E1	E2
Microcontroller	Arduino UNO R3	LilyGO TTGO V1.4
Battery Life	168 h	100 h
ADS1115 pressure transmitter	Not necessary	It is necessary
Fault prediction	Does not support	Supported
SIM800L module	For connection	Integrated
Power connector	USB-B	USB-C
Operating voltage	5 V DC	3.3 V DC
Difficulty of connection	Medium	Low
Control chip	ATmega 328p	Espressif ESP32-Wrover-B
Wi-Fi service connection	Does not support	Supported

**Table 4 sensors-22-03549-t004:** The irrigation machine technical data and other details.

Z	Technical Data						Weather Condition	
-	Type	L, m	D, mm	A, -	C, -	P, MPa	Ws	Te
							m.s^−1^	°C
Z1	Irtec 90 G/400	380	90	gun	E1	0.59 ± 0.02	1 ± 0.1	10 ÷ 40 (27.9)
Z2	Irtec 82 G/400	350	82	gun	E2	0.59± 0.01	1 ± 0.1	11 ÷ 44 (27.8)

Z—irrigation machine, L—length, D—diameter of hose, A—type of sprinkler, C—type of designed equipment, P—pressure, Te—temperature, Ws—wind speed.

**Table 5 sensors-22-03549-t005:** Output from the device in table form.

ID	SID	Datatype	Data	Timestamp
829	****** f64gsf ******	Distance [m]	0.00	2021-07-06 12:25:29
833	****** f64gsf ******	Distance [m]	352.00	2021-07-06 12:27:20
837	****** f64gsf ******	Distance [m]	352.00	2021-07-06 12:28:07
845	****** f64gsf ******	Distance [m]	352.00	2021-07-06 12:37:39



1409	****** f64gsf ******	Distance [m]	52.00	2021-07-07 03:43:39
1421	****** f64gsf ******	Distance [m]	49.50	2021-07-07 03:57:10
1445	****** f64gsf ******	Distance [m]	34.50	2021-07-07 04:47:21
-	Average values			

ID—Record number, Auto-incremental, SID—Device-specific number (star symbols serves as a protection of sensitive data–unique codes of already connected devices), Datatype—data type, for example position or temperature, Data—column of measured data itself, Timestamp—time stamp when the data were sent to the server.

## Data Availability

Not applicable.

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
