# Peer review of "Design of Hardware and Software Equipment for Monitoring Selected Operating Parameters of the Irrigator"

_sensors, 2022, doi:10.3390/s22093549_

Round 1
Reviewer 1 Report
The manuscript of sensors entitled Design of hardware and software equipment’s for monitoring selected operating parameters of the irrigator by Ján Jobbágy et al. design two devices for monitoring selected irrigation for two hose reel irrigation. It is hoped that the current research will support our modern agricultural irrigation system. The abstract is written well but last lines about conclusion add one more sentence is better.
The introduction in the manuscript a hose reel is mentioned for different crops vegetables and potatoes. Can this hose reel be used for major crops like cotton and wheat? What are the specific reasons for using a reel in a corn field?
The methodology of current research mention in the line 164 (The technical specifications are given in Table 1.). While, manuscript did not find table 1 technical specifications were found in table 2 please correct the mistake.
The author designed two devices (Arduino UNO and LilyGO TTGO V1.4 module) which device is more helpful for hose reel machines and, which is more beneficial in the irrigation field because according to table 3, E2 equipment was fewer hours required for battery, low voltages are required. Is there specific reason to select 10 mints time interval?
Please add a reference to the Results and Discussion line 705 (there are many publications). Further, 710, 711, please provide a reference because it’s strong statements and why not mention other countries?, and the same sentence is repeated in 724 and 725; this sentence is only written in a more fixed place in the manuscript
The writer stated in his conclusion that “The proposed facilities show that proper plant operation and timely response can help create more efficient and sustainable irrigation services, not only saving water but also reducing costs for the owner”. I was expecting to see some mathematical values presented in both conclusion and abstract part of the work. Some parts of the graph were not visible mostly, the headings. What are the research limitation of the study also provide the future research direction based on current research.

Author Response
Dear reviewer,
find attached file containing the responses from authors on your comments.
Best regards.

Reviewer 2 Report
The manuscript handles with the monitoring of operating parameters of the irrigator, although contains some questions.
The abstract is very long, more than 350 words. in the instructions for the author it has 200 as a maximum. Thus getting something scattered.
Line 123-133: please rephrase the goals, it’s very hard to understand, mainly this phrase: “The advantage of commercial systems is the photovoltaic panel, which is not yet in our facility.”. The rest of the text does not mention photovoltaic panels. After introduction, I cannot understand the goals of the work.
The link on line 588 does not work.
Author Response

(The authors gave the same response as above.)

Round 2
Reviewer 1 Report
The manuscript has been sufficiently improved to warrant publication.